# Factors Associated with the Duration of Breastfeeding: The Practices of Mexican Mothers in a Megacity and in the Agricultural Town

**DOI:** 10.3390/ijerph192215176

**Published:** 2022-11-17

**Authors:** Rocio Aidee Castillo-Cruz, Maria de la Luz Iracheta-Gerez, Mercedes Macias-Parra, Marcelino Esparza-Aguilar

**Affiliations:** 1Unit of Research in Epidemiology, Instituto Nacional de Pediatría, Av. Insurgentes Sur 3700-C Insurgentes Cuicuilco, Mexico City 04530, Mexico; 2Subdirectorate of Outpatient Care, Instituto Nacional de Pediatría, Av. Insurgentes Sur 3700-C Insurgentes Cuicuilco, Mexico City 04530, Mexico; 3Board of Directors, Instituto Nacional de Pediatría, Av. Insurgentes Sur 3700-C Insurgentes Cuicuilco, Mexico City 04530, Mexico

**Keywords:** breastfeeding, exclusive breastfeeding, complementary feeding, breastfeeding practices, infant formula, rooming-in care, skin-to-skin contact

## Abstract

Background: Breast milk is irreplaceable for healthy development. In Mexico, by 2019, the prevalence of exclusive breastfeeding (EBF) was low and the use of breastmilk substitutes (BMSs) was high. Objective: The aim of this work was to evaluate the maternal and child characteristics related to breastfeeding (BF) duration and to the introduction of BMSs for residents of Mexico City (CdMX) and an agricultural town in Morelos. Methods: A cross-sectional study was conducted with 160 mother–child binomials (0–15 months of age) from the megacity CdMX and the agricultural town. Outcomes: EBF and total breastfeeding (TBF) duration, age of transition to BMSs, and the introduction of complementary feeding (CF) were assessed. Associations with maternal and infant factors were assessed using Cox models. Results: The prevalence of EBF in the joint samples at 5.9 months was 32.6% and 5.8% at 6 months. EBF was favored under the following conditions: living in CdMX, receiving prenatal care, no newborn hospitalization, and breastmilk provided as first food at birth. TBF was prolonged under the following conditions: older mother, female children, rooming-in care during puerperium, receiving BF upon discharge after birth, cohabiting with extended family, and having no siblings. The introduction of BMSs predominated under the following conditions: living in an agricultural town, BMSs given after birth before discharge, younger mother, worker mother, and lack of prenatal care. The early introduction of CF (before the fourth month) was 2% for CdMX and 14% for the agricultural town. Conclusions: The agricultural population had a higher risk of the premature interruption of EBF/TBF and the early introduction of BMSs and CF. Protective factors were family-friendly environments and being born in a baby-friendly hospital.

## 1. Introduction

Breastfeeding impacts multiple aspects of the health and development of the mother and child including a reduction in the risk of breast and ovarian cancer and postpartum depression in the mother, a reduction in the risk of obesity and type 2 diabetes for the mother and child, and the opportunity for better cognitive development for the infant [1,2]. In 2012, the World Health Organization established the goal that by 2025, 50% of children under 6 months old will be exclusively breastfed [2]. Victora et al. assert that “the scaling up of breastfeeding to a near universal level could prevent 823,000 annual deaths in children younger than 5 years and 20,000 annual deaths from breast cancer” [3]. In Mexico, the prevalence of EBF according to the 2016 National Health and Nutrition Survey (Encuesta Nacional de Salud y Nutrición-ENSANUT, its acronym in Spanish) was 14.4%, while in the 2015 National Survey of Boys, Girls, and Women (Encuesta Nacional de Niños, Niñas y Mujeres-ENIM, its acronym in Spanish), the prevalence of EBF was estimated at 30.8%, with a participation rate in urban areas of 76.8% [4,5]. In Mexico, according to the 2018–2019 National Health and Nutrition Survey (ENSANUT 2018–2019), the prevalence of exclusive breastfeeding in children under six months was 28.3%, 42.9% of children under 12 months consumed breastmilk substitutes (infant formula), and 29% continued breastfeeding at two years. Around 30% of children between 6 and 11 months did not meet the recommended minimum dietary diversity [6].

In Mexico, childhood obesity is a public health concern. A study by Huh and Cols in 2011 provides evidence regarding the unfavorable synergism of two factors in the feeding history of infants: not having been breastfed or suspending breastfeeding before the fourth month of age and the introduction of complementary feeding (CF) before the fourth month of life were associated with a six-fold increase in the odds of obesity at 3 years [7]. Exclusive breastfeeding from the first hours of life to 6 months contributes to the health of the intestinal microbiota through to the microbiota–gut–brain axis. Recent research points to the role of unbalanced gut microbiota (dysbiosis) and the possible link to other problems of the nervous system such as autism spectrum disorder [8].

The objective of our study was to identify the factors associated with the duration of exclusive breastfeeding (EBF), total breastfeeding (TBF), and the introduction of complementary feeding (CF) during the first 15 months of age, using a survival method to model across time, and to describe the practices of Mexican mothers at two different places: a megacity (Mexico City, CdMX) in comparison with an agricultural town (Tlaltizapán Morelos, Mexico).

## 2. Materials and Methods

### 2.1. Design

This was a cross-sectional analytical study with 160 healthy mother–child pairs attending well-child care visits. After written informed consent was obtained, a face-to-face structured interview was carried out to apply a questionnaire lasting approximately 20 min in Spanish. Maternal, perinatal, and infant factors were studied. The main outcomes were duration of EBF and TBF according to the definition by the World Health Organization (WHO), age of initiation with breastmilk substitutes (BMSs), and CF. Associations with maternal, perinatal, and infant factors were determined using Cox models, reporting hazard ratios (HRs) with a 95% confidence interval (CI).

### 2.2. Participants and Settings

The research was conducted during 2016. Mothers (aged 15 to 41 years) of children (aged 0 to 15 months) attending well-child care visits at the National Institute of Pediatrics in the south of CdMX (*n* = 100) and at the Health Center in the municipality of Tlaltizapán at Morelos State (*n* = 60) were included. Classification of infants by outcome type: Respecting the definitions provided by the WHO, each child was classified as EBF, TBF, BMSs, or CF, and the evolution of the events of interest was represented with Kaplan–Meier curves and Cox proportional hazards models. We considered foods other than milk such as porridge of plant or animal origin, and CF.

Statistical analysis. The duration of EBF and TBF, and the age of the infant when BMSs and CF were introduced were calculated from birth to the moment (in months) when the event occurred or until the time of the interview to censor the cases without the presence of the event. EBF interruption was defined by the following three moments: introduction of BMSs, introduction of CF, or complete cessation of breastfeeding. The data for the participants from CdMX and the agricultural town were compared using Fisher’s exact test for categorical variables or the Wilcoxon rank-sum test for numerical variables. Bivariate associations were determined using the Cox proportional hazards test. It is worth noting that survival methods have been used to analyze cross-sectional data elsewhere when the aim was to analyze the outcome distribution over time [9,10,11,12,13]. The variables that were associated with a significance level <0.05 or with a tendency toward significance (<0.10) were tested in a multivariable Cox proportional hazards model. Microsoft ^®^ Office Excel^®^ 2007, Epi Info™ 7.2.0.1, and R version 3.4.3 were used [14,15,16,17].

### 2.3. Ethical Considerations

Written consent was obtained from all parents or guardians of children involved in the study and was approved by the Institutional Review Board of the National Institute of Pediatrics.

### 2.4. Data Collection

Data collection was performed by trained paramedic personnel through face-to-face interviews in which a clinical questionnaire was applied that included variables established through an advanced search of the international and national literature with terms related to characteristics in the study population regarding the start and interruption of breastfeeding. Data related to birth such as weight and length were obtained from the information contained in the national health card and somatometry from the day of the interview.

Principal independent variables were the place of residence, maternal age, history of prenatal care, child number, sex of the baby, complications during the obstetric event, skin-to-skin contact, presence of extended family in the home, and type of birth.

## 3. Results

### 3.1. Description of the Sample

Differences were found in the distribution of variables between participants from CdMX and the agricultural town. Mothers from CdMX were older (*p* = 0.0009), had a higher education level (*p* = 0.0019), worked more often (*p* = 0.0148), and were more often outside the home (*p* = 0.086), and among the mothers who worked, they did so for more hours (*p* = 0.0056), attended prenatal care slightly earlier (*p* = 0.0383), and reported a lower proportion of hospitalization related to the obstetric event (*p* = 0.0131). Although they were hospitalized, on average, for one more day because of the obstetric event (*p* = 0.0001), they reported a greater proportion of skin-to-skin contact (*p* = 0.0019) and a lower proportion of rooming-in care (*p* = 0.0075). The mothers started BMSs 29 days later (*p* = 0.0076) and indicated that the doctor was the one to guide the feeding of the child (*p* = 0.0002). At least three-quarters introduced CF after 6 months of age (somewhat later compared with the agricultural town, *p* = 0.0267) and reported the presence of disease more frequently (*p* = 0.0144). The characteristics of the participants are provided in Table 1 and Table 2.

The infants from CdMX were generally 2 months older at the time of the interview (*p* < 0.0001), and at the time of the survey, they had a shorter length-for-age (*p* = 0.0057). CF was introduced with a higher frequency of vegetables and fruits and a lower frequency of meat and legumes (*p* = 0.0147). A higher proportion of these infants were hospitalized at birth (*p* = 0.0025). A smaller proportion of these children were an only child (*p* = 0.0091), and therefore, had a greater number of siblings (*p* = 0.0338). There was less cohabitation with four or more non-first-degree relatives (*p* = 0.0165) (Table 1 and Table 2). There were no bivariate differences between groups in variables such as maternal marital status, history of prenatal care, history of receiving breastfeeding information during prenatal care and/or the obstetric event, type of birth (delivery or cesarean), place of birth, complications resulting from the obstetric event, anthropometric indicators at birth, duration of hospitalization of the child, first complementary food received, history of receiving BMSs at the birth center, food provided to the child upon discharge from the birth center, age when breastfeeding was suspended, weight and length for age at the time of the interview, breastfeeding status, use of BMSs and onset of CF at the time of the interview, and reasons for the indication of BMSs; the data are provided in Table 1 and Table 2.

#### 3.1.1. Exclusive Breastfeeding

The general prevalence of EBF in the two groups together was 32.6% at 5.9 months and 5.8% at 6 months of age by the introduction of BMSs or TBF. EBF lasted longer in the mother–child pairs from CdMX (*p* < 0.0001) than in the mother–child pairs from the agricultural town; from the first day of extrauterine life, EBF was 85.0% and 61.7%, respectively, whereas at one month, the percentages were 73.0% and 47.9%, respectively.

The percentage of EBF interruption before 3 months of age, considered as “early abandonment” [18], was 38% in CdMX and 57% in Tlaltizapán Morelos (i.e., was 19% higher for the agricultural town). At the fifth month, the percentages were 47.9% and 16.4%, respectively, and at the sixth month (i.e., the age at which CF is usually introduced), the percentages fell to 8.7% and 0.0%, respectively (HR: 0.4, 95% CI: 0.26–0.62 for the CdMX sample) (Figure 1A).

Based on bivariate analysis, other factors that favored EBF (i.e., were against its interruption) were higher maternal age (HR: 0.96, 95% CI: 0.92–0.99), prenatal care beginning in the first trimester of pregnancy (HR: 0.21 95% CI: 0.05–0.93), maternal nonhospitalization at birth (HR: 0.65, 95% CI: 0.43–0.98), nonhospitalization of the newborn (HR: 0.62, 95% CI: 0.40–0.99), rooming-in care in a bed or in a crib (HR: 0.49, 95% CI: 0.27–0.89), having breast milk as the first food at birth (HR: 0.51, 95% CI: 0.33–0.80), not receiving BMSs at the birth hospital (HR: 0.58, 95% CI: 0.33–0.89), and having siblings (HR: 0.59, 95% CI: 0.39–0.90). Conditions that showed a trend (*p* ≥ 0.05 and *p* < 0.1) of sustaining breastfeeding were a history of non-obstetric complications, initiation of prenatal care in the second trimester (compared to not having prenatal care), and receiving BMSs and maternal breast milk as food upon leaving the hospital (compared to receiving BMSs only). A history of concomitant dengue at birth (HR: 11.47, 95% CI: 1.07–123.29) was identified as a risk factor for EBF duration, and the maternal grandmother (as opposed to the physician) was the main figure considered for guiding feeding of the infant after leaving the hospital (HR: 2.3, 95% CI: 1.08–4.91) (Table 3 and Figure 1B–D).

In the multivariable analysis, the preserved factors associated with EBF duration were the population of CdMX (HR: 0.28, 95% CI: 0.17–0.46) and prenatal care (HR: 0.07, 95% CI: 0.02–0.34), nonhospitalization of the neonate (HR: 0.5, 95% CI: 0.3–0.84), and receiving maternal milk as the first food (HR: 0.45, 95% CI: 0.27–0.73), or even “other” food (HR: 0.16, 96% CI: 0.03–0.92), but not BMSs. In this model, infants having siblings trended toward significance as a protective factor (*p* = 0.07). (Table 4).

#### 3.1.2. Total Breastfeeding

The prevalence of TBF was not different between the mother–child pairs from CdMX and those from the agricultural town (*p* = 0.9829). On the first day of extrauterine life, the TBF prevalence was 94.0% and 96.6% for mother–child pairs from CdMX and the agricultural town, respectively; at the first month, the values were 92% and 94.5%; at 6 months, the values were 75.8% and 75.6%; and at one year, the values were 62.2% and 75.6%, respectively (Figure 1A).

In the bivariate analysis, TBF duration was associated with the infant being female (HR: 0.44, 95% CI: 0.21–0.88), no complications (maternal or neonate) at birth (HR: 0.36, 95% CI: 0.19–0.72), skin-to-skin contact at birth (HR: 0.47, 95% CI: 0.23–0.95), rooming-in care of the mother–child pair during the health service stay after birth (HR: 0.39, 95% CI: 0.18–0.86), not receiving BMSs during the postnatal health service stay (HR: 0.39, 95% CI: 0.18–0.82), receiving breast milk (and not BMSs) when leaving the hospital after birth (HR: 0.12, 95% CI: 0.05–0.28), and the child cohabitating with three or more second-degree or greater relatives (HR: 0.25, 95% CI: 0.08–0.84). Among the conditions at the time of birth that subsequently affected TBS interruption were complications prior to birth (oligohydramnios, cephalopelvic disproportion, alterations in labor, and premature rupture of membranes, HR: 4.14, 95% CI: 1.16–14.75), hypertensive disease (HR: 3.52, 95% CI: 1.44–8.62), and infants with birth weight <−2 standard deviations (HR: 6.11, 95% CI: 1.28–29.26). Protective factors with a tendency toward significance were receiving maternal breast milk as the first food (*p* = 0.06), and receiving maternal breast milk and BMSs, but not only BMSs, when leaving the hospital after birth (*p* = 0.08), as shown in Table 3 and in Figure 1E,F.

In the multivariable analysis, maternal age (in years, HR: 0.88, 95% CI: 0.81–0.95) remained significantly associated with TBF duration, in addition to the child being female (HR: 0.40, 95% CI: 0.17–0.92), rooming-in care after birth (HR: 0.34, 95% CI: 0.12–0.97), receiving maternal breast milk as food upon leaving the hospital after birth (HR: 0.03, 95% CI: 0.01 −0.11), or even milk and BMSs (HR: 0.09, 95% CI: 0.02–0.31) but not only BMSs, the child cohabitating with three or more second-degree or greater relatives (HR: 0.11, 95% CI: 0.03–0.46), and having no siblings (HR: 0.033, 95% CI: 0.14–0.76), as shown in Table 4.

#### 3.1.3. Introduction of Breastmilk Substitutes

On the first day of extrauterine life, 38% and 13% of the infants from the agricultural town and CdMX received BMSs, respectively; by the first month of life, 52% and 26% had received BMSs, respectively; and by the sixth month, 69% and 46% had received BMSs, respectively (Figure 1A). In a bivariate analysis, a delay in the introduction of BMSs was related to the population (HR: 0.49, 95% CI: 0.31–0.78 for CdMX population). Additionally, the delayed use of BMSs was related to prenatal care (HR: 0.22, 95% CI: 0.05–0.93), start of prenatal care in the first trimester of pregnancy (HR: 0.21, 95% CI: 0.05- 0.90), type of birth (HR: 0.61, 95% CI: 0.39–0.96 for vaginal delivery), no complications at birth (HR: 0.55, 95% CI: 0.33–0.90), receiving maternal breast milk as the first food at birth (HR: 0.49, 95% CI: 0.31–0.79), not receiving BMSs from the health service after birth (HR: 0.41, 95% CI: 0.25–0.66), receiving maternal breast milk upon discharge from a medical unit after birth (HR: 0.39, 95% CI: 0.21–0.75), and the child having siblings (HR: 0.61, 95% CI: 0.39–0.96). As significant risk factors for the introduction of BMSs, dengue at birth (HR: 10.95, 95% CI: 1.11–108.29), obstetric complications before birth (oligohydramnios, cephalopelvic disproportion, alterations in labor, and premature rupture of membranes, HR: 2.89, 95% CI: 1.11–7.47), and the maternal grandmother being the predominant figure guiding the feeding of the child upon leaving the hospital after birth (HR: 2.26, 95% CI: 10.6–4.48) were identified (Table 3).

In the multivariable analysis, the introduction of BMSs was delayed in mother–child pairs from CdMX (HR: 0.37, 95% CI: 0.22–0.62), with each year of maternal age (HR: 0.95, 95% CI: 0.90–0.99), with prenatal care (HR: 0.15, 95% CI: 0.03–0.73), and when BMSs were not received in the birth center (HR: 0.28, 95% CI: 0.16–0.47); additionally, one risk factor for introducing BMSs was a working mother (HR: 1.92, 95% CI: 1.07–3.46), as shown in (Figure 1G–I).

#### 3.1.4. Introduction of Complementary Feeding

The presence of CF at 2, 5, 6, and 12 months of age occurred in 2%, 15%, 81%, and 95%, respectively, of the mother–child pairs from the CdMX sample, and in 14%, 43%, 87%, and 100%, respectively, of the mother–child pairs from the agricultural town. The early introduction of CF, defined as CF introduction before the fourth month of life, was 2% for CdMX and 14% for Tlaltizapán.

In the bivariate analysis, an older age of CF introduction was related to the population (HR: 0.35, 95% CI: 0.18–0.71 for CdMX), to the maternal place of work being the home (HR: 6.65, 95% CI: 1.49–29.57), and to not receiving BMSs in the hospital after birth (HR: 2.33, 95% CI: 1.17–4.66). In addition, a length for age <−3 Z at the time of the interview was associated with an earlier CF introduction (HR: 10.42, 95% CI: 1.55–70.03). On the other hand, prenatal care through private health services, compared to public services, showed a trend toward significance for the later introduction of CF (*p* = 0.07), while birth by vaginal delivery and receiving maternal milk as food upon leaving the hospital after birth showed a trend toward significance for the earlier introduction of CF (*p* = 0.05 and *p* = 0.09, respectively) (Table 3).

In the multivariable analysis, the later introduction of CF was related to CdMX as the place of origin (HR: 0.35, 95% CI: 0.17–0.70), while an earlier introduction was related to mothers who worked within the home (HR: 6.88, 95% CI: 1.47–32.19) and to infants with length for age <−3 Z at the time of interview (HR: 7.84, 95% CI: 1.06–57.64). The complete model is presented in Table 4.

## 4. Discussion

Our findings are consistent with international reports with respect to the risk and protective factors for the initiation and length of breastfeeding, as reported by Ortega et al., in Murcia, Spain, in a prospective cohort as well as by Avila Ortiz in relation to the factors associated with abandoning EBF in Mexican mothers attending two private hospitals in a high socioeconomic status population where BMSs are introduced early by medical indication [19,20]. In this study, the factors with beneficial effects on breastfeeding were higher maternal age and education level as well as prenatal care, birth by delivery, family support, receiving training during pregnancy, not offering BMSs at the birth center, skin-to-skin contact, no neonatal complications, female child, and the presence of other children and extended family. Although the mother–child pairs included were based on convenience sampling, the results reflect the variation that exists in terms of EBF duration. Even the prevalence of breastfeeding was comparable to that described in the Spanish population by Oribe et al., where it was 15.4% for up to 6 months [21,22,23].

The agricultural town population (from a municipality with 52,399 inhabitants) was at risk of premature EBF/TBF interruption, a result that was not expected. Notably, the prevalence of breastfeeding at the end of 5 months (13.7%) for the agricultural population of Tlaltizapán Morelos was consistent with the estimate of 14% for children under 6 months by national surveys (ENSANUT 2012); notably, this value is overrepresented for households experiencing social deprivation [4,24]. In comparison, our sample from CdMX (a megacity with 22.1 million habitants) had an EBF percentage at the end of the fifth month of age of 42.1%, which was more comparable to the 2015 ENIM, reporting an EBF prevalence of 30.8% in children under 6 months of age.

In these samples, we identified clinical characteristics related to feeding practices that had previously been associated with a 6-fold increased risk of obesity at preschool age such as abandonment of breastfeeding before 3 months of age. The introduction of CF before the fourth month predominated in the agricultural population of Tlaltizapán, with 57% (14% for CdMX); ref. [7] in comparison, the early abandonment of EBF in children under 3 months and the introduction of CF before the fourth month were 38% and 0% in CdMX.

One of the benefits of knowing the risk factors in particular populations is the identification, during the prenatal stage, of mother–child pairs that are at risk, which in turn has the potential to facilitate both intervention with the already tested strategies and the investigation of the efficacy of new specific proposals [25].

There is a Cochrane review that can be used to update the recommendations of evidence-based breastfeeding. The aim of this review was to describe the forms of breastfeeding support that have been assessed in controlled studies, the timing of interventions, and the settings in which they have been used [26]. The authors searched for evidence from February 2016; the 73 trials contributed to the analyses were from 29 countries and involved 74,656 women. Some 62% of the women were from high-income countries, 34% from middle-income countries, and 4% from low-income countries.

The methodological quality of the studies was mixed and the components of the standard care interventions and extra support interventions varied greatly, and were not always well described. The authors found that additional support from both lay people and professionals had a positive impact on breastfeeding outcomes.

Our study provides information on the identification of possible risk and protective factors with respect to pre- and postnatal breastfeeding practices, some statistically significant and others with clinical relevance despite the limited sample, within two different groups with respect to socioeconomic and cultural conditions in the context of Mexico, currently the country with the second largest population in Latin America.

Breastfeeding support can change our understanding of how best to help women continue with EFB and TBF until their child is two years of age or older [26], based on the perspective of the role that breastfeeding plays as a protective factor against breast cancer in young women [27]. However, this is a problem with multiple dimensions, ranging from the medical context and personal preferences of women, to the social, economic, and health policy issues that consist of creating programs and providing program supervision, market regulations, and prescriptions for milk substitutes, as noted by the National Academy of Medicine of Mexico in 2016 [28,29]. We agree that breastfeeding can help bridge the gap between rich and poor, thus reducing inequity [3,30].

We found that the survival model was a useful approach to describe the practices of breastfeeding in Mexican mothers and to compare two different places with statistical and clinical significance.

## 5. Limitations

Our Kaplan–Meier curves and Cox proportional hazards models were based on reconstructions of the evolution of the events of interest recorded in a cross-sectional measurement, not a true follow-up, and may therefore be subject to memory bias; however, we did not qualitatively identify any factor or subgroup where such potential measurement error could lead to poor differential classification. On the other hand, this potential measurement error is inherent to cross-sectional measurement and would still be present if any other method of analysis had been used. The current analysis is an effort to overcome the temporal ambiguity of cross-sectional studies to offer a more dynamic description of breastfeeding; more importantly, the EBF prevalence in the agricultural population at 5 months of age is consistent with estimates for a comparable age in the 2012 ENSANUT and ENSANUT 2018–2019 data, which overrepresent populations such as that of Tlaltizapán Morelos [6,22]. As the data collection began in 2016, and several social and health events have happened since (such as the COVID-19 pandemic), the actual figures concerning the practices and trends in breastfeeding in the studied populations could have changed; however, the associated factors found here would surely still have an effect.

## 6. Conclusions

The prevalence of EBF was low in the study groups. The factors that favor EBF duration in this study were living in the megacity of Mexico, the relatively older age of the mother, a history of prenatal care, and the child not being an only child.

The factors that favored TBF duration were early (in the first months of pregnancy) prenatal medical check-ups, a female child, obstetric history without complications, skin-to-skin contact, and the presence of extended family in the home (three or more non-first-degree relatives).

The predisposing factors for initiating BMSs were belonging to the agricultural population of Tlaltizapán Morelos, younger infant age, not attending prenatal care, birth by cesarean section, history of obstetric complications, not starting CF at the time of interview, and being an only child.

The age of introduction of solid foods or CF was influenced by the type of population, with an earlier introduction in the agricultural population of Morelos.

Cases at risk of various unfavorable outcomes for breastfeeding may be identifiable from the prenatal stage. It is necessary to conduct deeper research on these factors to accurately and prenatally identify those subjects at greater risk, and to effectively implement a national breastfeeding strategy for the promotion, monitoring, and disclosure of favorable practices for successful breastfeeding, encouraging births in a baby-friendly hospital.

## Figures and Tables

**Figure 1 ijerph-19-15176-f001:**
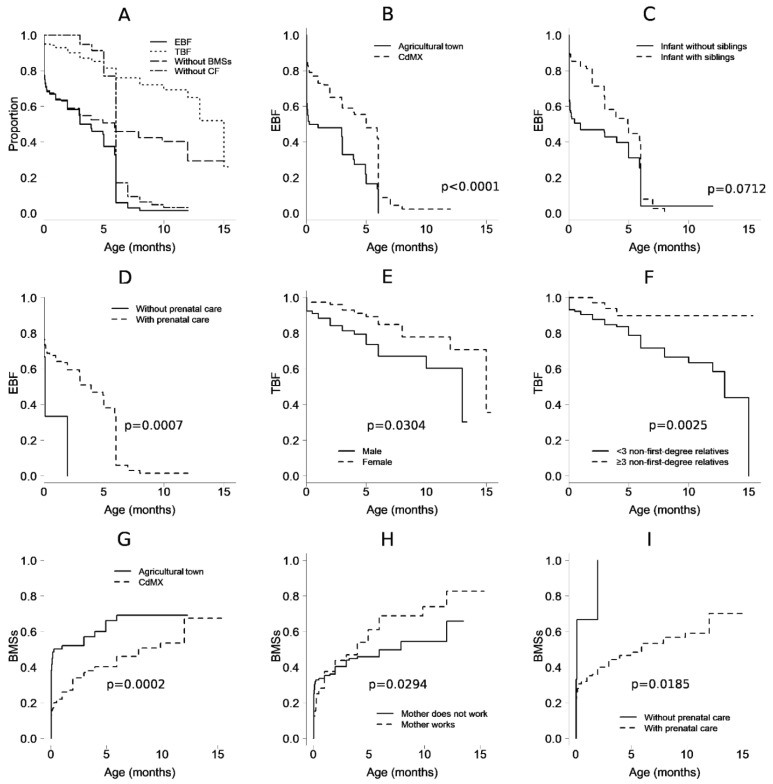
Observed proportion over time, in months, from birth, of infants who received (**A**) exclusive breastfeeding (EBF) and total breastfeeding (TBF, i.e., exclusive or nonexclusive) and who did not receive milk substitutes (BMSs) or complementary feeding (CF). (**B**) EBF in mother–child pairs from Mexico City and from an agricultural town in Morelos. (**C**) EBF and the presence of siblings. (**D**) EBF and prenatal care. (**E**) TBF and sex. (**F**) TBF and the presence of extended family in the home. (**G**) BMSs and the Mexico City or Morelos populations. (**H**) BMSs and maternal work conditions. (**I**) BMSs and prenatal care conditions. The *p*-values correspond to multivariable models.

**Table 1 ijerph-19-15176-t001:** Description of the sample from Mexico City and Tlaltizapán, Morelos. Numerical variables.

Variables	Mexico City			Tlaltizapán			*p* Value Ws-r
	Median	Q1	Q3	Median	Q1	Q3	
Mother’s age (years)	26	21	31	22	19	25	0.0009
Mother’s education (years)	12	9	12	9	9	12	0.0019
Mother’s hours of work (working mothers)	9	7	10	2	1	4	0.0056
Mother’s hours of work (all mothers)	0	0	3	0	0	0	0.0007
Days worked by the mother (working mothers)	5	5	6	5	5	7	0.8785
Days worked by the mother (all mothers)	0	0	2	0	0	0	0.0162
First month of pregnancy in which the mother went to the doctor	2	1	3	2	2	3	0.0383
Mother’s duration of hospitalization (days)	2	2	3	1	1	2	0.0001
Child’s duration of hospitalization (days)	3	2	5	2	1	3	0.3604
Duration of skin-to-skin contact (minutes)	2	1	5	10	4	20	<0.0001
Child’s age at interview (months)	6	5	10	4	1	7	<0.0001
Age in months at breastfeeding interruption (those who have already been interrupted)	5	3	8	4	3	5	0.2190
Age in months at breastfeeding interruption (all)	3	1	6	3	1	4	0.3966
Age at onset of BMSs use, in days	30	1	96	1	1	5	0.0076
Age at onset of CF (months)	6	6	6	6	5	6	0.0267
Number of siblings	1	0	1	0	0	1	0.0338
Weight/height at birth, percentile	17.9	4.5	44.0	12.5	1.5	46.4	0.7005
Height/age at birth, percentile	67.7	31.9	83.9	67.7	31.9	86.9	0.7121
Weight/age at birth, percentile	34.8	19.1	53.2	36.9	13.8	59.5	0.9956
Current weight/height, percentile	50.4	29.1	81.2	45.2	10.2	72.2	0.0566
Current height/age, percentile	18.1	5.9	33.5	40.5	10.4	82.6	0.0057
Current weight/age, percentile	34.1	13.0	55.4	37.4	11.5	62.9	0.6025

Ws-r: Wilcoxon signed-ranks test. BMSs: breast milk substitutes. CF: complementary feeding.

**Table 2 ijerph-19-15176-t002:** Description of the sample from Mexico City and Tlaltizapán, Morelos. Nominal variables.

Variables/Category		Mexico City	Tlaltizapán	*p* Value
		%	%	Fisher’s Exact Test
Marital status	Married	25.0	16.7	0.4721
	Single	13.0	15.0	
	Consensual union	62.0	68.3	
Working mother		26.0	10.0	0.0148
Place of work	Home	4.0	5.0	0.0086
	Away from home	22.0	5.0	
Attended prenatal medical care		97.0	100.0	0.2925
Type of prenatal care service	Public	80.0	90.0	0.4052
	Private	13.0	8.3	
	Other	4.0	1.7	
Received breastfeeding information prenatally		86.9	73.3	0.0741
Child’s sex	Male	52.0	45.0	0.4178
Type of birth	Vaginal delivery	57.0	60.0	0.7429
Birth place	A hospital	98.0	100.0	0.9999
	Home	1.0	0.0	
	Public	1.0	0.0	
Complications at birth		25.0	20.0	0.5627
Mother’s hospitalization		49.0	70.0	0.0131
Child’s hospitalization		33.0	11.7	0.0025
Skin to skin contact		39.0	65.0	0.0019
Rooming-in care	In bed	18.0	86.7	0.0075
	In crib	67.0	3.3	
	Without rooming-in.	15.0	10.0	
First food that was given to the child	Maternal milk	63.0	66.7	0.8558
	BMSs	33.0	28.3	
	Other #	2.0	1.7	
	Do not know	2.0	3.3	
Received BMSs (infant formula) in the hospital	Yes	54.0	51.7	0.2858
	No #	39.0	46.7	
	Do not know	7.0	1.7	
What did the child take when leaving the hospital?	Maternal milk	72.0	76.7	0.8318
	Infant formula	13.0	11.7	
	Both	15.0	11.7	
Who indicated what was taken when leaving the hospital?	Doctor	81.0	65.0	0.0002
	Nurse	11.0	10.0	
	Maternal grandmother	1.0	21.7	
	Paternal grandmother	2.0	1.7	
	Other	5.0	1.7	
Currently, the child breastfeeds		73.0	83.3	0.1753
Currently, the child receives BMSs		50.5	50.0	1.0000
Currently, the child initiated CF		57.0	51.7	0.5171
First complementary food	Meat, legumes, broth	4.0	11.7	0.0147
	Cereals	0.0	3.3	
	Vegetables and fruits	52.0	36.7	
	Other	1.0	0.0	
The child has siblings		55.0	33.3	0.0091
The child cohabitates with four to six extended family members		8.0	21.7	0.0165
Mother with disease		9.0	0.0	0.0144
Weight to length at birth < −2 Z		15.0	26.7%	0.0821
Weight to height at the current time < −2 Z		1.0	6.7	0.0599

BMSs: breast milk substitutes. CF: complementary feeding. Z: standarized Z-score. # In Mexico, some infant feeding practices (not recommended but popular) include infusions, cow milk, corn-based beverages, etc.

**Table 3 ijerph-19-15176-t003:** Bivariate analysis between EBF, TBF, BMSs, and CF with sociodemographic and maternal and child factors.

Background/Exposure		EBF Interruption HR (95% CI, *p* Value)	TBF Interruption HR (95% CI, *p* Value)	BMSs Introduction HR (95% CI, *p* Value)	CF Introduction HR (95% CI, *p* Value)
CdMX (reference: agricultural town)		**0.4** **(0.26–0.62, <0.0001)**	1.19 (0.55–2.6, 0.6568)	**0.49** **(0.31–0.78, 0.0029)**	**0.35** **(0.18–0.71, 0.0031)**
Maternal age (years)		**0.96** **(0.92–0.99, 0.0215)**	0.96 (0.9–1.02, 0.1829)	0.96 (0.92–1, 0.0579)	0.98 (0.93–1.04, 0.5262)
Maternal schooling (years)		0.98 (0.92–1.04, 0.5045)	0.98 (0.88–1.09, 0.6851)	1.01 (0.94–1.08, 0.8402)	0.93 (0.85–1.03, 0.1675)
Mother works (reference: does not work)		1.11 (0.68-1.81, 0.6781)	1.27 (0.6-2.67, 0.5365)	1.38 (0.83–2.3, 0.2186)	0.97 (0.48–1.94, 0.9296)
Where mother works (reference: does not work)	Home	1.66 (0.66–4.14, 0.2781)	1.06 (0.14–8.14, 0.9538)	1.6 (0.59–4.31, 0.3534)	**6.65** **(1.49–29.57, 0.0129)**
	Away from home	1 (0.58–1.72, 0.9968)	1.29 (0.59–2.83, 0.5164)	1.33 (0.76–2.33, 0.3221)	0.75 (0.36–1.58, 0.4478)
With prenatal care (reference: without prenatal care)		**0.22** **(0.05–0.95, 0.0426)**	0.29 (0.03–2.38, 0.2488)	**0.22** **(0.05–0.93, 0.0397)**	1.32 (0.07–25.0, 0.8534)
Trimester of initiation of prenatal care (reference: without prenatal care)	1st	**0.21** **(0.05–0.93, 0.0404)**	0.25 (0.03–2.11, 0.2033)	**0.21** **(0.05–0.9, 0.0349)**	1.42 (0.07–27.15, 0.8158)
	2nd	0.23 (0.05–1.11, 0.0675)	0.44 (0.05–4.02, 0.4636)	0.27 (0.06–1.26, 0.0948)	1 (0.05–20.42, 0.9985)
	3rd	0.2 (0.02–2.04, 0.1756)	0 (0–Inf, 0.9973)	0.21 (0.02–2.7, 0.233)	1.23 (0.02–91.31, 0.9248)
Type of medical care service (reference: public)	Private	1.24 (0.65–2.38, 0.5142)	1.95 (0.83–4.6, 0.1275)	1.87 (0.99–3.54, 0.0532)	0.4 (0.15–1.07, 0.0691)
	Other	1.81 (0.59–5.54, 0.2949)	0 (0–Inf, 0.9967)	1.45 (0.42–5.07, 0.5572)	1.38 (0.29–6.67, 0.6862)
	Without prenatal care	**4.8** **(1.1–20.98, 0.037)**	3.72 (0.45–31.07, 0.2248)	**5.02** **(1.17–21.51, 0.0296)**	0.69 (0.04–13.38, 0.8088)
Child’s sex (reference: male)		0.9 (0.6–1.36, 0.622)	**0.44** **(0.21–0.88, 0.0216)**	0.8 (0.51–1.26, 0.3382)	1.22 (0.67–2.25, 0.5183)
Vaginal delivery (reference: cesarean section)		0.77 (0.51–1.16, 0.207)	0.66 (0.33–1.29, 0.2219)	**0.61** **(0.39–0.96, 0.0342)**	1.92 (0.99–3.72, 0.0531)
No obstetric complications (reference: complications)		0.67 (0.42–1.06, 0.0892)	**0.36** **(0.19–0.72, 0.0034)**	**0.55** **(0.33–0.9, 0.017)**	1.61 (0.8–3.25, 0.1832)
Dengue at birth (reference: without dengue)		**11.47** **(1.07–123.29, 0.0441)**	2.76 (0.34–22.18, 0.3399)	**10.95** **(1.11–108.29, 0.0407)**	0.94 (0.1–8.88, 0.9547)
Obstetric complications (reference: no complications)	Oligohydramnios, CPD, alteration in labor, PROM	1.93 (0.72–5.19, 0.1922)	**4.14** **(1.16–14.75, 0.0286)**	**2.89** **(1.11–7.47, 0.029)**	0.56 (0.15–2.15, 0.3985)
	Tear, atony or uterine rupture, ileus	1.96 (0.56–6.91, 0.2936)	2.06 (0.26–16.27, 0.4933)	2.54 (0.67–9.57, 0.168)	0.46 (0.07–3.21, 0.4356)
	Hypertensive disease	1.47 (0.73–2.94, 0.2775)	**3.52** **(1.44–8.62, 0.0058)**	1.42 (0.67–3.02, 0.363)	0.72 (0.24–2.22, 0.5705)
	Other comorbidities	1.27 (0.45–3.64, 0.6505)	2.21 (0.49–9.88, 0.3002)	1.62 (0.54–4.88, 0.3898)	1.27 (0.3–5.4, 0.7428)
	Fetal distress, meconium aspiration	1.06 (0.31–3.64, 0.9274)	1.28 (0.26–6.28, 0.7615)	1.51 (0.43–5.33, 0.5203)	0.28 (0.05–1.58, 0.1499)
No maternal hospitalization (reference: inpatient maternity)		**0.65** **(0.43–0.98, 0.0404)**	0.79 (0.4–1.56, 0.4997)	0.67 (0.42–1.05, 0.083)	0.85 (0.46–1.57, 0.6054)
No newborn hospitalization (reference: newborn hospitalization)		**0.62** **(0.4–0.99, 0.0436)**	0.71 (0.34–1.46, 0.3491)	0.64 (0.39–1.05, 0.0766)	1.22 (0.61–2.46, 0.5688)
With skin-to-skin contact (reference: without skin-to-skin contact)		1.0 (0.67–1.5, 0.999)	**0.47** **(0.23–0.95, 0.036)**	0.91 (0.58–1.43, 0.6838)	1.41 (0.75–2.65, 0.2904)
Rooming-in in bed or on a cot		**0.49** **(0.27–0.89, 0.0189)**	**0.39** **(0.18–0.86, 0.0203)**	0.59 (0.31–1.1, 0.0971)	1.61 (0.66–3.93, 0.2944)
First food at birth (reference: BMSs)	Breast milk	**0.51** **(0.33–0.8, 0.003)**	0.53 (0.27–1.03, 0.0614)	**0.49** **(0.31–0.79, 0.0034)**	1.7 (0.83–3.45, 0.144)
	Other	0.24 (0.04–1.35, 0.1056)	0.0 (0–Inf, 0.9975)	0.66 (0.14–3.06, 0.5946)	0.15 (0.02–1.41, 0.0973)
	Do not know	0.6 (0.13–2.73, 0.5047)	0.0 (0–Inf, 0.998)	0.72 (0.15–3.35, 0.6746)	0.0 (0–Inf, 0.9972)
BMSs in the hospital (reference: receiving BMSs in the hospital)	No BMSs in the hospital	**0.58** **(0.38–0.89, 0.0124)**	**0.39** **(0.18–0.82, 0.0128)**	**0.41** **(0.25–0.66, 0.0003)**	**2.33** **(1.17–4.66, 0.0162)**
	Do not know	0.59 (0.21–1.65, 0.3164)	0 (0–Inf, 0.9962)	0.59 (0.2–1.71, 0.3301)	2.96 (0.68–12.91, 0.1495)
Food upon leaving the hospital (reference: BMSs)	Breast milk	**0.37** **(0.2–0.7, 0.0023)**	**0.12** **(0.05–0.28, <0.0001)**	**0.39** **(0.21–0.75, 0.0046)**	2.5 (0.88–7.07, 0.0854)
	Both	0.5 (0.22–1.11, 0.0898)	0.42 (0.16–1.1, 0.0775)	0.52 (0.22–1.21, 0.1293)	2.03 (0.59–7.05, 0.2632)
Person who indicated (reference: doctor)	Maternal grandmother	**2.3** **(1.08–4.91, 0.0314)**	1.94 (0.65–5.76, 0.2355)	**2.26** **(1.06–4.84, 0.0359)**	1.35 (0.36–5.08, 0.6534)
	Paternal grandmother	0.54 (0.1–3.06, 0.489)	2.07 (0.26–16.34, 0.4898)	0.46 (0.06–3.53, 0.4558)	0.44 (0.03–5.81, 0.5297)
	Nurses	0.91 (0.46–1.8, 0.7831)	0.93 (0.28–3.13, 0.9042)	0.9 (0.41–1.96, 0.7927)	1.2 (0.44–3.31, 0.7244)
	Other	1.47 (0.56–3.84, 0.4304)	1.69 (0.38–7.43, 0.4865)	1.63 (0.61–4.38, 0.3293)	2.06 (0.35–12.3, 0.428)
With siblings (reference: without siblings)		**0.59** **(0.39–0.90, 0.0141)**	1.51 (0.75–3.03, 0.2233)	**0.61** **(0.39–0.96, 0.034)**	0.83 (0.45–1.54, 0.55)
Three or more extended relatives (excludes first-degree)		1.19 (0.75–1.89, 0.4661)	**0.25** **(0.08–0.84, 0.0241)**	1.03 (0.61–1.72, 0.9222)	1.46 (0.68–3.15, 0.3309)
Weight to age at birth < −2 Z		1.59 (0.34–7.55, 0.5568)	**6.11** **(1.28–29.26, 0.0236)**	1.61 (0.34–7.56, 0.5441)	0 (0–Inf, 0.9975)
Length to age at interview < −3 Z		1.92 (0.53–6.88, 0.3184)	1.49 (0.19–11.54, 0.7009)	0.53 (0.07–4.07, 0.5423)	**10.42** **(1.55–70.0, 0.0159)**

EBF: exclusive breastfeeding. TBF: total breastfeeding. BMSs: breast milk substitutes. CF: complementary feeding. HR: hazard ratio. CI: confidence interval. CdMX: Mexico City. CPD: cephalopelvic disproportion. PROM: premature rupture of membranes. Bold text corresponds to the HRs of significant factors.

**Table 4 ijerph-19-15176-t004:** Factors associated with the interruption of EBF and TBF and the introduction of BMSs and CFs using multivariable proportional hazards models.

Outcome	Factor/Category		HR (CI 95%, *p* Value)
EBF interruption (LR test (7 df) *p* < 0.0001 (*n* = 160, events = 129))	Urban population		0.28 (0.17–0.46, <0.0001)
	With prenatal control		0.07 (0.02–0.34, 0.0007)
	No newborn hospitalization		0.5 (0.30–0.84, 0.0082)
	First food at birth (reference: BMSs)	Maternal milk	0.45 (0.27–0.73, 0.0012)
		Other	0.16 (0.03–0.92, 0.0397)
		Do not know	0.67 (0.13–3.32, 0.6213)
	With siblings		0.67 (0.43–1.04, 0.0712)
TBF interruption (LR test (7 df), *p* < 0.000 (*n* = 159, events = 36))	Maternal age (in years)		0.88 (0.81–0.95, 0.0021)
	Child’s gender		0.4 (0.17–0.92, 0.0304)
	Rooming-in care		0.34 (0.12–0.97, 0.0444)
	Food after hospital discharge (reference: BMSs)	Maternal milk	0.03 (0.01–0.11, <0.0001)
		Both	0.09 (0.02–0.31, 0.0002)
	With siblings		0.33 (1.32–7.16, 0.0092)
	Cohabitates with more than three relatives of degree greater than the first		0.11 (0.03–0.46, 0.0025)
BMSs introduction (LR test (6 df) *p* < 0.0001 (*n* = 160, events = 87))	Urban (reference: agricultural town)		0.37 (0.22–0.62, 0.0002)
	Maternal age (for each year)		0.95 (0.9–0.99, 0.0153)
	Mother without paid work (reference: paid)		0.52 (0.29–0.93, 0.0294)
	With prenatal care (reference: without prenatal care		0.15 (0.03–0.73, 0.0185)
	BMSs in the hospital (reference: received BMSs in the hospital	Did not receive BMSs in hospital	0.28 (0.16–0.47, <0.0001)
		Do not know	0.63 (0.21–1.89, 0.4146)
CF introduction (LR test (4 df) *p* = 0.0019 (*n* = 159, events = 88))	Urban population		0.41 (0.2–0.83, 0.0129)
	Place of work (reference: does not work)	In the home	6.88 (1.47–32.19, 0.0143)
		Away from home	0.86 (0.4–1.85, 0.6931)
	Length to age at the time of the interview < −3 Z		7.84 (1.06–57.69, 0.0432)

EBF: exclusive breastfeeding. TBF: total breastfeeding. BMSs: breast milk substitutes. CF: complementary feeding. HR: hazard ratio. CI: confidence interval. LR test: likelihood ratio test. df: degrees of freedom.

## Data Availability

The data presented in this study are available on request from the corresponding author.

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
