# Peer review of "Factors Associated with the Duration of Breastfeeding: The Practices of Mexican Mothers in a Megacity and in the Agricultural Town"

_ijerph, 2022, doi:10.3390/ijerph192215176_

Round 1

Reviewer 1 Report (Previous Reviewer 1)

no further comment, but some English editing may be needed for readability.

Author Response

We appreciate your kind review of the article. The version with style correction carried out by the professional service offered by the journal's publisher is sent..

Reply: It was already done and we sent the version edited by the MDPI translation service.

On the other hand, although there is no specific comment regarding the evaluation of the study design in the sense that it could be improved, we want to express that in the previous review we have taken into account the comments of the reviewers, the limitations of the aspects of the design and analysis that can no longer be modified have been accepted and made explicit, but we are aware that this small cross-sectional study with positive results (consistent with the literature on the subject) will serve as the basis for providing the assumptions of a more robust sampling, both for cross-sectional and longitudinal studies.  Thank you

This manuscript is a resubmission of an earlier submission. The following is a list of the peer review reports and author responses from that submission.

Round 1

Reviewer 1 Report

In the first reading was a bit puzzling as to what the objectives of the study were. The description that the sample was from a cross-sectional, convenient sampling, with unknown method on sample size estimation, hence, rather inadequate to answer the intended objectives.

Reading through, however, it appears that the statistical approaches are interesting, and would better contribute to the field. If agreed, the manuscript needs to be totally revised with appropriate background and rationale to justify the analytical approach used. The rest of the manuscript should also be revised and present the statistical approach as the key content, while the data used can be declared as a convenient sample from two locations, etc., and used to demonstrate the analytical approach.  Lastly, the discussion should bring out the key feature of the paper, i.e., analytical approach used in this study, advantages, etc., compared to other more commonly used statistical methods (e.g., multiple linear regression or logistic regression). 

Prevalence and factors associated with breastfeeding (BF) practices in Mexico city and an agricultural village. The topic itself is not original, as there has been similar studies reported in nutrition literature, although from different countries/settings.  

Despite of the previous comments, I found the analytical approach is of interest and not frequently used in the studies I have seen. Hence, if shown to be statistical appropriate, this paper could contribute to the field. The use of convenient sample and a relatively small sample size could be argued for this to be a demonstration on the use of the method.

In the present manuscript, objective is unclear whether it is intended to compare prevalence and factors on breastfeeding practices between the two locations. There is no sample size estimation method whether for prevalence estimate or for assessing associations between outcomes and independent variables. There was very limited information about the ample, beyond indicated that the data are cross-sectional from two locations: one is a mega-city and another, an agricultural village. If the main objective is to assess the prevalence and associated factors of the whole sample, then what the sample represents becomes an issue as well. Moreover, for the analysis which have over 20 explanatory factors but using a sample size of 160, is this adequate to answer the intended questions need discussion on how the results be interpreted.

While authors did make conclusion about the findings from the present study, although it is a bit unusual to state that the findings are similar to other previous studies, instead of bringing out what this study specifically contributes. The recommendation made in the final paragraph – it is not clear what ‘validations of these models’ that the authors refer to. The final sentence also made an extension to be used for public health promotion, surveillance, etc.

Several references are in Spanish; many are also (unpublished) reports which are available from websites

Table 1 may not be appropriate, given the size of the sample, and somewhat providing the same information as Table 2.

Figure 2 – what are the basis for selecting which outcomes and explanatory variables are to be presented (almost all were statistical significant, except for Figure 1 (C). Are these all bivariate analysis? In addition, to be consistent, there is one outcome missing, i.e. ‘introduction of complementary feeding’ which are not analyzed in a similar manner to the three outcomes.

Author Response

Reviewer 1

In the first reading was a bit puzzling as to what the objectives of the study were. The description that the sample was from a cross-sectional, convenient sampling, with unknown method on sample size estimation, hence, rather inadequate to answer the intended objectives.

Replay: There are several methodological and logistical details, and even results, that remained not mentioned in order to keep the report in a moderate extension. But, as there were significant positive associations, sample size in this study had the enough statistical power to detect the intended variable relations and reach the study objectives.

Reading through, however, it appears that the statistical approaches are interesting, and would better contribute to the field.

If agreed, the manuscript needs to be totally revised with appropriate background and rationale to justify the analytical approach used. The rest of the manuscript should also be revised and present the statistical approach as the key content, while the data used can be declared as a convenient sample from two locations, etc., and used to demonstrate the analytical approach.

Replay: We thank the suggestion, but that approach could be not necessary, and is out of the aim of our study. Survival methods are well known and stablished statistical procedures, and have been used to analyze cross-sectional data as in wide range of health topics as in:

  • Wang,R.; Tulikangas, P.K.; Tunitsky-Bitton, E. Relationship Between Maternal Age at First Delivery and Subsequent Pelvic Organ Prolapse. Female Pelvic Med Reconstr Surg. 2021,27,e423-e426. doi: 10.1097/SPV.0000000000000952.
  • Santibanez, T.A.; Srivastav, A.; Zhai,Y; Singleton, J.A. Trends in Childhood Influenza Vaccination Coverage, United States, 2012-2019. Public Health Rep. 2020,135,640-649. doi: 10.1177/0033354920944867
  • Knoll,P.; Rai, S; Talluri, S; Bezinque, A.; Micciche, R; Rao,G; Ankem,M.K. A Survey of Usage of Penile Prosthesis. J Sex Med. 2020,17,2287-2290. doi: 10.1016/j.jsxm.2020.07.082.
  • Lazzeri, G.; Tosti, C.; Pammolli, A.; Troiano, G.; Vieno, A.; Canale, N.; Dalmasso, P.; Lemma, P.; Borraccino, A.; Petraglia, F.; Luisi, S. Overweight and lower age at menarche: evidence from the Italian HBSC cross-sectional survey. BMC Womens Health. 2018,18, 168.
  • Gadkari, A.S.; Pedan, A.; Gowda, N.; McHorney, C.A. Survey nonresponders to a medication-beliefs survey have worse adherence and persistence to chronic medications compared with survey responders. Med Care. 2011,49, 956-61. doi: 10.1097/MLR.0b013e3182204503.

We included the text "It is worth noting that survival methods have been used to analyze cross-sectional data elsewhere when the aim was to analyze the outcome distribution over time" and included these references in the Methods/Analysis section.

Lastly, the discussion should bring out the key feature of the paper, i.e., analytical approach used in this study, advantages, etc., compared to other more commonly used statistical methods (e.g., multiple linear regression or logistic regression).

Replay: We thank the suggestion, but that approach could be not necessary as the analysis method depends mainly on the scales or levels of measurement of the dependent variables (linear regression for numerical, logistic for categorical binomial and proportional hazard for binomial with time of occurrence), and those statistical tests cannot be compared in terms of advantages as if were interchangeable.

Prevalence and factors associated with breastfeeding (BF) practices in Mexico City and an agricultural village. The topic itself is not original, as there has been similar studies reported in nutrition literature, although from different countries/settings.

Reply: We know that there is great amount of literature on the topic but, as breastfeeding practices are changing everywhere and every time, and as they are highly dependent on the context, there is a need for their study at those particular contexts where differences and inequities are problematized. Also, up to our knowledge, reports analyzing the effects of potential explanatory factors over the time to the outcomes are scarce.

Despite of the previous comments, I found the analytical approach is of interest and not frequently used in the studies I have seen. Hence, if shown to be statistical appropriate, this paper could contribute to the field. The use of convenient sample and a relatively small sample size could be argued for this to be a demonstration on the use of the method.

Reply: As mentioned, a statistical demonstration would be out of the objective (and maybe out of the journal scope), and sample size and sampling are commented elsewhere.

In the present manuscript, objective is unclear whether it is intended to compare prevalence and factors on breastfeeding practices between the two locations. There is no sample size estimation method whether for prevalence estimate or for assessing associations between outcomes and independent variables. There was very limited information about the ample, beyond indicated that the data are cross-sectional from two locations: one is a mega-city and another, an agricultural village. If the main objective is to assess the prevalence and associated factors of the whole sample, then what the sample represents becomes an issue as well. Moreover, for the analysis which have over 20 explanatory factors but using a sample size of 160, is this adequate to answer the intended questions need discussion on how the results be interpreted.

Reply: As it is stated, each group was formed by a convenient sample which, as in every non-random sample, represents itself, and that does neither prevent the description of summary statistics as group prevalence nor the search for associations. Also, inherent to the non-random nature of the samples, the interpretation of results is referred just to the studied groups/individuals, given that those results are not intended to represent to a wider population or sampling frame. We think that the sample prevalences and the other summary statistics are necessary to quantify magnitudes, to describe groups and to properly carry out the analysis as in other quantitative studies.

While authors did make conclusion about the findings from the present study, although it is a bit unusual to state that the findings are similar to other previous studies, instead of bringing out what this study specifically contributes. The recommendation made in the final paragraph – it is not clear what ‘validations of these models’ that the authors refer to. The final sentence also made an extension to be used for public health promotion, surveillance, etc. Several references are in Spanish; many are also (unpublished) reports which are available from websites.

Reply: the statement of similarity to other studies is to highlight consistency with those reports, even when statistical methods are different, but the related factor have analogy. Our contributions are considering the time to event and the consistent related factors described for the studied populations. References in Spanish have the corresponding English title. Some references were updated and those in Spanish actually do not have an English version.

Table 1 may not be appropriate, given the size of the sample, and somewhat providing the same information as Table 2.

Reply: Tables 1 and 2 are the conventional initial description of the groups/samples including the variables (numeric in table 1 and nominal in table 2) that entered to inferential analysis. Some variables could be related, but are not the same information, for example, for weight to height indicator, the distribution of the percentile value (numerical) is reported with no differences between groups, but the clinical classification (undernutrition, binary) had a trend towards significance in the difference between groups.

Figure 2 – what are the basis for selecting which outcomes and explanatory variables are to be presented (almost all were statistically significant, except for Figure 1 (C). Are these all bivariate analysis? In addition, to be consistent, there is one outcome missing, i.e. ‘introduction of complementary feeding’ which are not analyzed in a similar manner to the three outcomes.

Reply: In the figure 1 (A) all outcomes are those studied including without complementary feeding (CF), and explanatory variables are just those we had space and wanted to illustrate because of differences between groups, we have not an algorithm or rule to select them. The curves correspond to the observed data, and the p values to the estimation of that factor in the multivariable analysis, ad that was specified in the figure.

Reviewer 2 Report

Thank you for this article on factors associated with early cessation of exclusive and total breastfeeding. I have a few recommendations to make this article stronger and clearer:

Introduction:

-I am confused by the numbers in lines 38-42--how are 40% EBF at 4 months, but 60% EBF at 6 months?

-Who set the goal in 2012 to improve EBF to 50% by 2025?

-The last few sentences need to be reorganized as I am confused how funding (for what programs) would save maternal and infant lives.

-throughout the article, there is a mismatch in when numbers are used versus spelled out; general convention is less than 10, write (e.g., eight) and greater than 10, use number (e.g., 12).

-Sentence in line 52 needs a period

-"through to the microbiota-gut-brain axis" is referring to the following sentence information, I believe (lines 60-62)

-the objectives should be clearer and numbered: "...were to 1. describe...2. describe maternal and child..."

Materials and Methods:

-The design of the study needs more information. For example, were interviews conducted all in Spanish? How long were the interviews? What sampling technique was used?

-Participants and Setting: This section needs information about the locations of the study (demographics) as well as about the participants (age range of mothers and infants).

Line 93 should begin a new section called "analysis," which should be moved after data collection

Results:

-what sort of diseases were mothers suffering from?

Line 152 on pg 5 needs the cities mentioned before "respectively"

Line 191 "for multivariate analysis" instead of "multivaraible"

"in the .... analysis" is used a good amount in this section; redundancy should be avoided

The first two paragraphs under 3.1.5 are confusing as written

Discussion:

-delete "A" before "our findings"

citations are needed for the first sentence of this section

-lines 283-284, do you mean Spanish-speaking? or Spanish?

-missing a discussion of what could account for the variations found between the two locations

The arguments in the last paragraph of this section should be expanded and articulated, as it jumps from a Cochrane review (first mention) to breastfeeding as a means to "bridge the gap" of inquities

Author Response

Reviewer 2

Introduction:

-I am confused by the numbers in lines 38-42--how are 40% EBF at 4 months, but 60% EBF at 6 months?

Replay: The statement was erased.

-Who set the goal in 2012 to improve EBF to 50% by 2025?

Replay: The World Health Organization was specified in the text with the corresponding bibliographic reference.

-The last few sentences need to be reorganized as I am confused how funding (for what programs) would save maternal and infant lives.

Replay: The sentence " If approximately 4.70 dollars per infant is invested, the lives of 823,000 children under 5 years of age and 20,000 women could be saved annually, representing one of the most cost-effective investments" was removed, and a citation was included (“the scaling up of breastfeeding to a near universal level could prevent 823,000 annual deaths in children younger than 5 years and 20,000 annual deaths from breast cancer”).

-throughout the article, there is a mismatch in when numbers are used versus spelled out; general convention is less than 10, write (e.g., eight) and greater than 10, use number (e.g., 12).

Replay: expressions for age such as "X months", "X years", and "type 2 diabetes" remained as originally spelled. Expression as "2 factors", "3 moments", "4 to 6 extended family members", "4 or more non-first-degree relatives", "3 relatives", "3 or more second degree" changed.

-Sentence in line 52 needs a period Replay: period before new paragraph included.

-"through to the microbiota-gut-brain axis" is referring to the following sentence information, I believe (lines 60-62) Reply: that phrase was moved to the end of the paragraph.

-the objectives should be clearer and numbered: "...were to 1. describe...2. describe maternal and child..." Replay: The new writing is "The objective of our study was to identify the factors associated with the duration of exclusive breast-feeding (EBF), total breastfeeding (TBF) and the introduction of complementary feeding (CF) during the first 15 months of age, using a survival method to model in the time, and to describe the practices of the Mexican mothers at two different places: a megacity (México City) in comparison with an agricultural town (Tlaltizapán Morelos, México)".

Materials and Methods:

-The design of the study needs more information. For example, were interviews conducted all in Spanish? How long were the interviews? What sampling technique was used?

Replay: The requested information was included. "After a written informed consent was obtained, a face to face structured interview was carried out to apply a questionnaire lasting about 20 minutes in Spanish language."

-Participants and Setting: This section need information about the locations of the study (demographics) as well as about the participants (age range of mothers and infants).

Replay: The participant's ages were included and the text "Mothers (aged 15 to 41 years) of children (aged 0 to 15 months) attending well-child care visits at the National Institute of Pediatrics in the south of Mexico City (n=100) and at the Health Center in the municipality of Tlaltizapán at Morelos State (n=60)" was moved to this section.

Line 93 should begin a new section called "analysis," which should be moved after data collection. Replay: Done; a "Statistical analysis" paragraph was completed.

Results:

-what sort of diseases were mothers suffering from?

Reply: Neither mothers nor children were selected because any disease, and their diagnostics statuses were not recorded. As stated, they were included because of attendance to medical follow-up consultations for their healthy children.

Line 152 on pgs. 5 needs the cities mentioned before "respectively".

Replay: the new wording is "The general prevalence of EBF in the two groups together was 32.6% at 5.9 months and 5.8% at 6 months of age by the introduction of BMSs or TBF".

Line 191 "for multivariate analysis" instead of "multivariable" Replay: changed to "multivariable"

"in the .... analysis" is used a good amount in this section; redundancy should be avoided

Replay: "In the multiple regression analysis" or "In the multivariable regression" changed to " In the multivariable analysis".

The first two paragraphs under 3.1.5 are confusing as written. Replay: New wording "The presence of CF at 2, 5, 6 and 12 months of age occurred in 2%, 15%, 81% and 95%, respectively, of the mother-child pairs from CdMX sample, and in 14%, 43%, 87% and 100%, respectively of the mother-child pairs from the agricultural town. The early intro-duction of CF, defined as CF introduction before the 4th month of life, was 2% for CdMX and 14% for Tlaltizapán.

In the bivariate analysis, an older age of CF introduction was related to the population (HR: 0.35, 95% CI: 0.18-0.71 for CdMX), to the maternal place of work being the home (HR: 6.65, 95% CI: 1.49-29.57) and to not receiving BMSs in the hospital after birth (HR: 2.33, 95% CI: 1.17-4.66). In addition, a length for age <-3 SD at the time of the interview was associated with an earlier CF introduction (HR: 10.42, 95% CI: 1.55-70.03). On the other hand, prenatal care through private health services, compared to public services, showed a trend toward significance for later introduction of CF (p=0.07), while birth by vaginal de-livery and receiving maternal milk as food upon leaving the hospital after birth showed a trend toward significance for the earlier introduction of CF (p=0.05 and p=0.09, respectively). "

Discussion:

-delete "A" before "our findings" Replay: Done.

citations are needed for the first sentence of this section

-lines 283-284, do you mean Spanish-speaking? or Spanish? Replay: yes, it refers to a population in Spain.

-missing a discussion of what could account for the variations found between the two locations

Replay: thanks,  the discussion was expanded on the topic.

The arguments in the last paragraph of this section should be expanded and articulated, as it jumps from a Cochrane review (first mention) to breastfeeding as a means to "bridge the gap" of inequities.

Replay:  thanks, the discussion was expanded on the topic.

Reviewer 3 Report

Dear authors, I congratulate you for your work.

Here are some suggestions after reviewing the manuscript:

Title:

Line 2 “duration and interruption”. I consider that these terms are synonyms in this particular case.

Abstract:

The study design is not specified in the title or abstract.

Lines 21 and 22: “The prevalence of EBF at 5.9 and 6 months of age were 32.6% and 5.8%, respectively.” This sentence is not understood: What is the prevalence of urban mothers and rural mothers? Each percentage must be specified to which participants it belongs.

Introduction:

Lines 66 to 68: “maternal and child characteristics, perinatal characteristics and social factors related to the duration and interruption of breastfeeding, as well as the age of the infant when CF was introduced, were evaluated through a direct interview.” I believe that this information should go in the “methods” section.

Lines 68 to 70: “We found associations with factors also noted in other studies that can be considered in future predictive models to identify pairs at risk of breastfeeding abandonment” These information are study results. Consider its inclusion in the “results” or “discussion” section.

Methods:

The data collection is indicated to have been carried out in 2016.

I consider that it should be included as a limitation of the study that the data collection was carried out 6 years ago, in 2016.

Results:

Lines 152 and 153: Specify here also the populations to which the percentages belong (urban vs. agrarian).

Author Response

Reviewer 3

Title:

Line 2 “duration and interruption”. I consider that these terms are synonyms in this particular case. Replay: The title changed to " Factors Associated with the Duration of Breastfeeding. The Practices of Mexican Mothers in a Megacity and in the Agricultural Town".

Abstract:

The study design is not specified in the title or abstract. Replay: The "cross-sectional" design was included in the abstract.

Lines 21 and 22: “The prevalence of EBF at 5.9 and 6 months of age were 32.6% and 5.8%, respectively.” This sentence is not understood: What is the prevalence of urban mothers and rural mothers? Each percentage must be specified to which participants it belongs. Replay: We specified that "The prevalence of EBF in the joint samples".

Introduction:

Lines 66 to 68: “maternal and child characteristics, perinatal characteristics and social factors related to the duration and interruption of breastfeeding, as well as the age of the infant when CF was introduced, were evaluated through a direct interview.” I believe that this information should go in the “methods” section. Replay:  Thanks. Text removed.

Lines 68 to 70: “We found associations with factors also noted in other studies that can be considered in future predictive models to identify pairs at risk of breastfeeding abandonment” This information are study results. Consider its inclusion in the “results” or “discussion” section. Replay: Thanks Text removed.

Methods:

The data collection is indicated to have been carried out in 2016. I consider that it should be included as a limitation of the study that the data collection was carried out 6 years ago, in 2016. Replay: Replay It was stated as a limitation

Results:

Lines 152 and 153: Specify here also the populations to which the percentages belong (urban vs. agrarian). Replay: It was specified that that figure corresponds to the pooled samples, and also the result for every group was specified.